# BRAIN-LIKE REPLAY NATURALLY EMERGES IN REINFORCEMENT LEARNING AGENTS

## ABSTRACT

Replay is a powerful strategy to promote learning in artificial intelligence and the brain. However, the conditions to generate it and its functional advantages have not been fully recognized. In this study, we develop a modular reinforcement learning model that could generate replay. We prove that replay generated in this way helps complete the task. We also analyze the information contained in the representation and provide a mechanism for how replay makes a difference. Our design avoids complex assumptions and enables replay to emerge naturally within a task-optimized paradigm. Our model also reproduces key phenomena observed in biological agents. This research explores the structural biases in modular ANN to generate replay and its potential utility in developing efficient RL.

## 1 INTRODUCTION

Experience replay (Lin, 1992) is an important technique for online reinforcement learning (RL). By storing past experiences in a circular buffer and training the agent on these experiences repeatedly, experience replay improves sample efficiency and training stability (Schaul et al., 2015), and successfully improves the performance of RL algorithms (Mnih, 2013; Mnih et al., 2015). There are several works on how to design more sophisticated experience replay mechanisms, like hyperparameter tuning (Fedus et al., 2020), prioritizing replay with higher temporal difference (TD) error (Schaul et al., 2015), or developing a generative replay model (Van de Ven & Tolias, 2018). However, given the significant efficiency gap between biological and RL agents, we explore whether replay algorithms can be further improved by drawing inspiration from brain structures.

In neuroscience, replay is the reactivation of a sequence of place cells that encode recently experienced locations. It has been observed in many brain areas during rest or sleep, from the hippocampus (Buzsáki, 1986; Nádasdy et al., 1999) to the neocortex, including the prefrontal cortex (Peyrache et al., 2009; Kaefer et al., 2020), visual cortex (Ji & Wilson, 2007), and motor cortex (Eichenlaub et al., 2020). Some people believe it represents the "virtual trajectory" in the brain that primarily serves two functions: memory consolidation (Foster & Wilson, 2006; O'Neill et al., 2010) and planning (Pfeiffer & Foster, 2013; Widloski & Foster, 2022). Its dynamic properties and functional diversity have attracted significant attention in neuroscience research.

There are currently two focuses in this field: the emerging condition of replay and the function of replay. Many models have been proposed to explain these two aspects. Some biophysical models suggest that replay arises from the combination of past experiences and specific neuronal network connectivity rules, such as asymmetrical synaptic connections (Tsodyks et al., 1996) and short-term depression (Romani & Tsodyks, 2015). In addition to connectivity rules, task optimization provides another way to understand the emergence of replay. (Krishna et al., 2024) proved that denoising dynamics under task optimization could lead to offline reactivation of past experiences. Levenstein et al. (2024) verified that only the network trained on multi-step predictive learning with recurrent connection and head-direction input can generate offline trajectories resembling replay. These models are illustrative in that they connect conditions like circuit structures, task optimization or predictive learning to the generation of replay. However, they fail to explain the functional importance of replay in the task.

On the other hand, experience replay provides good evidence that replay does have some importance and helps learning. However, this approach is not ideal for exploring the conditions under which replay emerges, as all replay of stored experiences is hard-coded and predefined. The model never

finds a way to organize experiences and perform efficient replay without force. The gap between these two research fields leads to the natural question of whether we could find some conditions under which a model could naturally generate adaptive replay sequences with functional importance. Identifying such conditions could provide important insights on how to design new reinforcement learning agents that quickly adapt to the environment, given the present low efficiency of RL algorithms and the unique advantage of biological agents in learning new tasks.

Recently, some studies have attempted to approach this problem within the RL framework. For example, Mattar & Daw (2018) propose that a special variable called the expected value of backup (EVB) determines the replay priority of different experiences, and the replay is accompanied by Bellman updates which are essential for learning the value function and policy for the task. However, because we cannot know EVB until we perform value iteration for the whole environment, and after value iteration it seems unnecessary to use replay at all, the generating condition in there are still some hard-coded constraints in this method.

In this article, we propose a model to circumvent hard-coded settings, provide some conditions as a natural explanation for the emergence of replay, and then show the functional importance of our generated replay sequences. In summary,

1. Based on two straightforward conditions, we develop a model capable of generating offline replay activation.

2. Throughout the entire learning process of the flexible navigation task, the evolution of replay spatial distribution of our model resembles that observed in real animals.

3. Through ablation experiments, we verify the functionality of replay by showing that the exploration efficiency of our model surpasses that of models without replay.

4. We analyze the information that replay flow carries, and explain how replay can promote efficient learning.

## 2 Methods

**Conditions to generate instrumental replay** What could these conditions possibly be? The main areas where replay occurs are the prefrontal cortex (PFC) and hippocampal formation (HF), usually modeled as policy network and world model respectively due to their unique functions. Replay in these two individual areas is often observed to be coordinated ("phase-lock") when a new environment learning task is presented (Place et al., 2016). We hypothesize that communication between these two areas is crucial for the generation of instrumental replay. Therefore, we propose two conditions that could be implemented in RL agents:

$$\text{Condition 1: Replay serves for reward maximization.} \quad (1)$$
$$\text{Condition 2: Replay is accompanied by communication between PFC and HF.}$$

To achieve *Condition 1*, we employ the end-to-end RL framework without hard-coded design. For *Condition 2*, our RL agent has a world model and a policy model, representing HF and PFC respectively, and we add an additional information passage between these two models that is trained alongside the policy network. This setting allows the agent to adjust the information communication flow based on its influence on actions and expected rewards. Discussions about the plausibility of these settings can be found in section 4.1.

**Principle of designing different modules** We preface the introduction of model structure with key biological biases that we incorporate into our design.

Biologically, HF has abundant recurrent connections in its CA3 area (Le Duigou et al., 2014), which is also thought to be a major area generating replay. Thus we implement a gated recurrent unit (GRU) for HF. Besides, there are two main functions for HF. First, the HF is good at encoding the environment structure (i.e. cognitive map) which can serve as a basis for path integration and flexible navigation (Whittington et al., 2018). In our setting, the HF module learns to do this by predicting the next location (path integration) and reward, based on history (hidden state) and environmental information (input) (Figure 1A). The HF is also thought to store episodic memory (Whittington et al., 2018; Battaglia et al., 2011), and in our model, it learns to memorize past reward history.

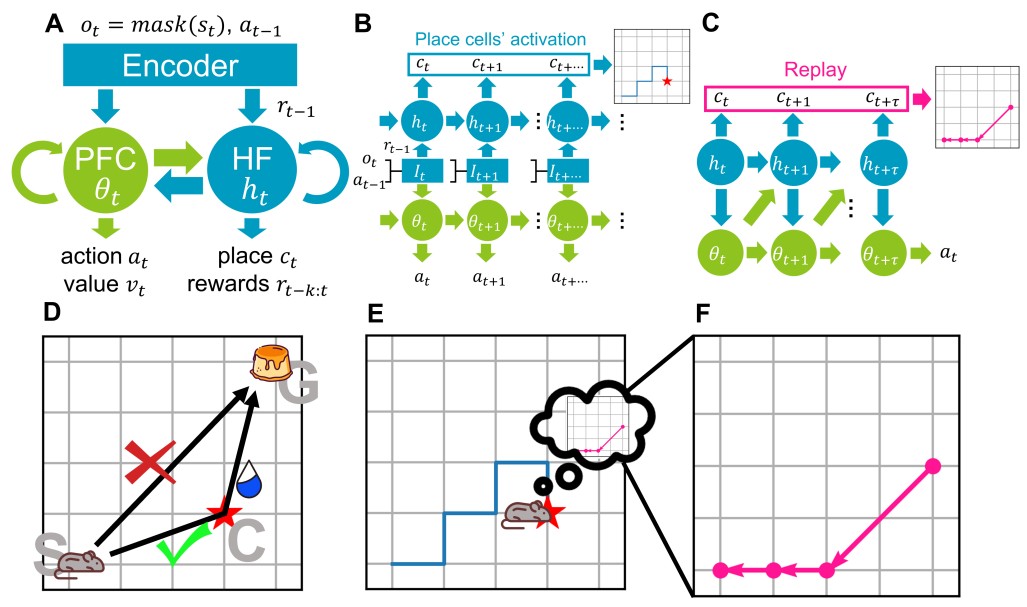

Figure 1: **A**-**C**. Model structure. **A** The HF GRU module completes the task of path integration and episodic memory. The PFC RNN module is responsible for decision-making. The information passage between the PFC and HF only opens at rest. **B**. During mobility, the information passage remains closed, and the HF and PFC modules operate independently. The place cell activation reflects the real present locations. **C**. During immobility, the information passage opens. The HF and PFC modules start to communicate with each other. The activated place cell output constituted a replay sequence. **D**-**F**. Task setting. **D**. The agent should start from S, first get to checkpoint C, consume a small amount of reward (0.5), and then arrive at goal G to get a large reward (1.0). Directly moving to G will cause no rewards. **E**. A representative trajectory generated by the trained RL agent. First, it reaches C and gets the small reward. The replay happens at this time. **F**. The replay event in **E**.

The PFC is also modeled as a recurrent network (Figure 1A) due to the recurrent nature of the real prefrontal cortex. The prefrontal cortex is thought to be the center for decision-making (Barra-clough et al., 2004) and value encoding (Padoa-Schioppa & Assad, 2006), and in our task, it learns to estimate values and select actions to maximize the expected reward using the hidden state and environmental information.

To handle high-dimensional visual inputs, we use an ancillary Encoder module modeled by a convolutional neural network (CNN), simulating the sensory cortex (Figure 1A). Regarding replay, we establish an information passage between HF and PFC. Because replay is only detected when the subject receives a reward and stops to consume it (Igata et al., 2021), the information passage remains closed during movement (Figure 1B) and opens when the agent receives a reward (Figure 1C). During the agent's movement, the PFC decides the agent's next action (i.e., up, down, left, or right) based on the input of environmental representation processed by the Encoder module. At this time, it does not communicate with the HF module. Meanwhile, the HF maintains its own dynamics for correct place cell activation and recent episodic memory (Figure 1B). When the agent receives a reward, the two modules interact for several steps and influence each other through the information passage. The reactivation of place cells in the HF is then probed and analyzed as replay.

**Design of the HF Module** The update formula of the HF module can be written as:

$$h_t = f_{\text{HF}}(W_h h_{t-1} + (1 - \mathbb{I}_{\text{replay}})W_{h,\text{in}}[o_t, r_{t-1}, a_{t-1}] + \mathbb{I}_{\text{replay}}W_{h,\text{r}}\theta_{t-1}), \qquad (2)$$

$$[\hat{r}_{t-k:t}, \hat{G}(s = 1 : m)_t] = \sigma(W_{h,\text{out}}h_t). \qquad (3)$$

$h$ is the hidden state of HF. Input consists of $o_t$, $r_{t-1}$, and $a_{t-1}$, which are the current partially observable state, the previous reward and the previous action, respectively. $W_h$, $W_{h,\text{r}}$, $W_{h,\text{in}}$, and

$W_{h,\text{out}}$ are trainable parameters of neural networks. $\theta$ is the hidden state of the PFC. $\mathbb{I}_{\text{replay}}$ indicates the beginning of the rest period when replay and communication happens and input stops. $f_{\text{HF}}$ is the activation function for the HF module. The HF module then anticipates a Gaussian distribution peaked at the next location as well as rewards for recent steps $\hat{r}_{t-k:t}$. $m = 5 \times 5$ is the number of locations in this grid world. $k = 10$ is the length of the reward history the agent needs to remember.

The loss for structural learning can be written as the cross-entropy between the softmax output, $\hat{G}(s)$), and the real distribution, $G(s)$: $\mathcal{L}_{\text{pred}-\text{loc},t} = -\sum_{s=1}^{m} \hat{G}(s) \log G(s)$. To equip the HF module with the reward structure of the environment, it predicts the reward of the next step, $r_t$, with the loss written as $\mathcal{L}_{\text{pred}-\text{r},t} = |\hat{r}_t - r_t|$. The loss of memorizing rewards is calculated as the L1 difference between the remembered rewards and the actual rewards: $\mathcal{L}_{\text{memory}-\text{r},t} = \sum_{i=1}^{k} |\hat{r}_{t-i} - r_{t-i}|$.

**Design of the PFC Module** The update formula of PFC could be written as

$$\theta_t = f_{\text{PFC}}(W_\theta \theta_{t-1} + (1 - \mathbb{I}_{\text{replay}})W_{\theta,\text{in}}[o_t, r_{t-1}, a_{t-1}] + \mathbb{I}_{\text{replay}}W_{\theta,\text{r}}h_{t-1}) \tag{4}$$

$$[a_t, v_t] = W_{\theta,\text{out}}\theta_t. \tag{5}$$

$\theta$ is the hidden state of the PFC module. Input $[o_t, r_{t-1}, a_{t-1}]$ and the indicator $\mathbb{I}_{\text{replay}}$ are the same as in Equation 2. $W_\theta$, $W_{\theta,\text{r}}$, $W_{\theta,\text{in}}$, and $W_{\theta,\text{out}}$ are trainable parameters of neural networks. $f_{\text{PFC}}$ is an activation function. The state value $v_t$ is also estimated by the PFC for training purposes. We can see that replay is optimized because it changes the hidden state $\theta_t$ of the PFC and thus affects the expected reward.

**Training and Test Paradigm** We first pre-train the world model HF and the Encoder using random trajectories. Then the weights of the Encoder and HF are frozen, and we start training the PFC module by proximal policy optimization (PPO) to maximize the reward. We conduct the following analysis with all model weights fixed.

The reward settings during pre-training and training are the same. The reward is randomly chosen from one of the nine locations in the center of the room, and the location is reset at a probability = 0.1 so that memorizing reward locations is helpful to finish the task. However, in the test stage, the reward is the same as in the biological experiment.

## 3 RESULTS

### 3.1 BIOLOGICALLY SIMILAR REPLAY SEQUENCES EMERGE AFTER TRAINING

**Task setting** We use the animal experiment from (Igata et al., 2021) as a reference, in which a rodent was trying to navigate to a dynamically changing reward location in an open area (Figure 1D). First, the animal learned to run from the starting point S to checkpoint 1 (C1) to get a small reward, and then run from C1 to G to get a large reward. Violation of the order would cause no return at all (Figure 1D). When the stage of pre-learning was finished, the small reward was relocated to checkpoint 2 (C2, Figure 2B), requiring the animal to learn a new optimal path (S-C2-G). This was the beginning of the learning stage. Initially, the animal continued to take the path S-C1-G followed by G-C2-G after failing to receive a reward at G. It then realized the existence of a shortcut (S-C2) and gradually settled on it. The aim was to investigate changes in the relative amount of replay sequences representing different parts of the room ("replay distribution" in this paper) during the learning stage when the animal was trying to adapt to the change of reward location. They found that the amount of replay of the original trajectory (S-C1, C1-G) decayed rapidly, while the shortcut (S-C2) increased even before the agent adopted it (Figure 2C). This result could be evidence that replay supports flexible navigation by exploring possible paths and strengthening better ones.

**Simulation result** We create a similar simulation environment as in (Igata et al., 2021). The animal is replaced by an RL agent moving in a $5 \times 5$ open arena. As in the animal experiment, we put the small reward at C1 and then relocate it to C2. Without adjusting the network parameters, and simply by modifying the hidden states of the RNNs, the RL agent successfully identifies the optimal path to the new checkpoint. (Figure 3A).

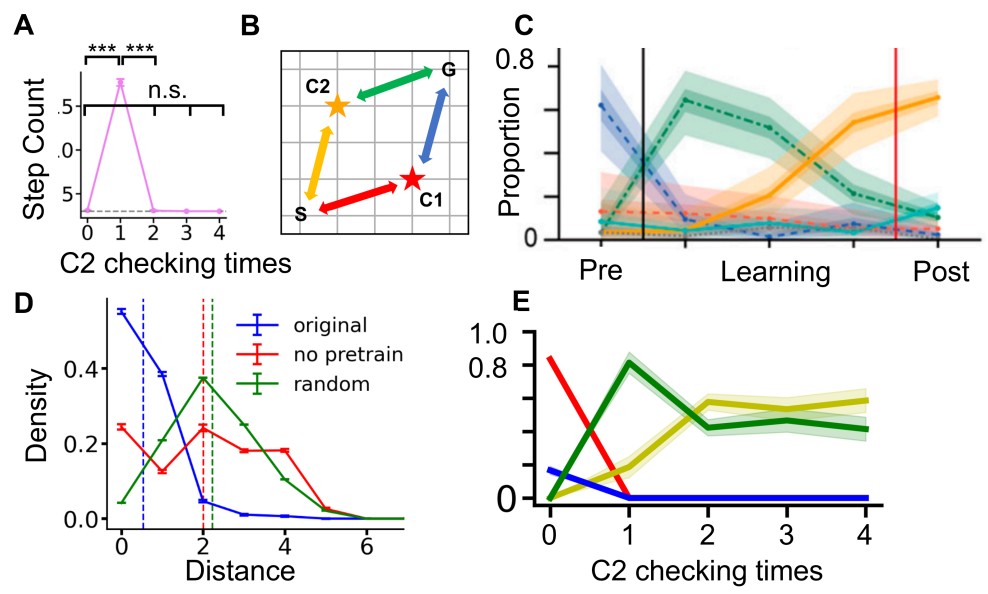

Figure 2: **A**. Performance for the RL agent in the test period. **B**. Legends for **C** and **E**. Different colors represent different parts of the room. **C**. Change of replay distribution for different segments in the animal data, adapted from (Igata et al., 2021). **D**. Distribution of distances between adjacent replay steps. **E**. Change of replay distribution generated by the RL agent.

The locations of C1 and C2 are the same as in the original experiment (Figure 2B). We first measure the distribution of distances of adjacent replay steps (Figure 2D). Compared with the random level or the model without pretraining, the optimized model shows a significant tendency to generate adjacent replay steps. Therefore, it tends to generate continuous trajectories rather than random skipping points.

We attribute each replay trajectory to one of the four paths in Figure 2B and then calculate the relative fraction of the four paths at each time step, as the "replay distribution" (See subsection A.1 for calculation details). The change in replay distribution of the RL agent (Figure 2E) closely mirrors that of the rodent (Figure 2C) in two important aspects. First, the amount of replay of C2-G increases and then drops down. Second, the amount of replay of S-C2 increases. Recalling that the animal first took path S-C1-G-C2-G and then S-C2-G, This trend possibly reflects the animal's focus from "finding a plausible path" to "finding an optimal path". These results prove that the conditions proposed in 1 are sufficient to generate replay.

### 3.2 ABLATION STUDY DEMONSTRATES THAT REPLAY HELPS LEARNING

To prove replay is functionally important, first, we replace the signal in the information passage to be random noise or an all-zero vector. Because the information exchange between HF and PFC is bidirectional, we carry out this replacement in both directions respectively and find that the performance is only impaired when the information from HF to PFC is replaced by meaningless signals. The information in the reverse direction is of no use (Figure 3A). This interesting finding echoes some neuroscientific discoveries that brain wave in HF leads that in PFC in the task encoding stage when phase lock between these two brain areas happens (Jadhav et al., 2016; Spellman et al., 2015).

Second, we replace the multi-step information exchange process with one-step information emission and retrain the RL part, i.e. the PFC network and information passage (Figure 3C, left). We measure the number of exploration steps it takes to get to the reward at the time when it's relocated but the agent continues to follow the old path S-C1 and fails to find it. The original model demonstrates a significantly lower number of exploration steps and higher exploration efficiency compared to the ablation models (Figure 3C, right). This suggests that multi-step information emission is superior to single-step emission. We further mask varying numbers of replay steps from the final decision

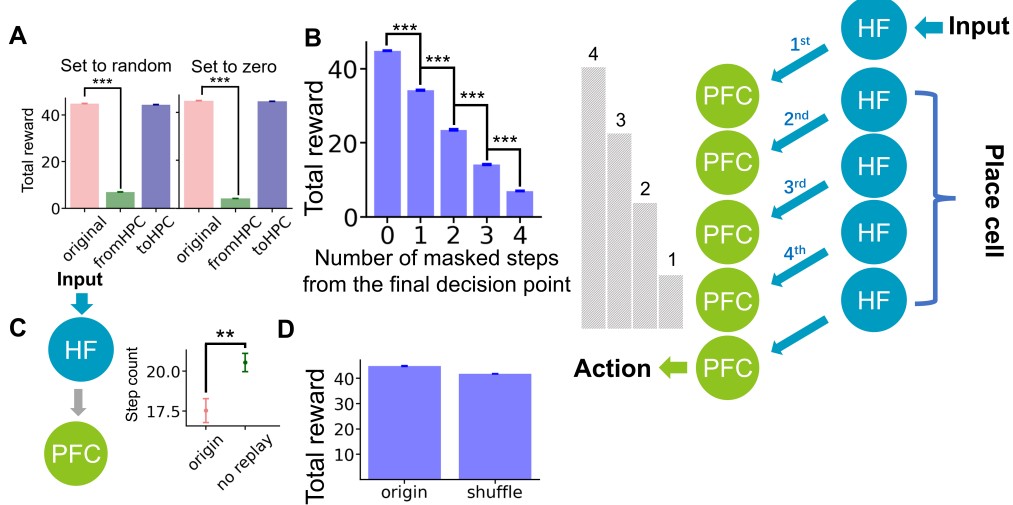

Figure 3: **A**. The total reward decreases when the signal from HF to PFC is replaced by random noise (left) or all-zero vectors (right). **B**. The different number of replay steps is masked (right), and the performance decreases monotonically as we mask more steps (left). **C**. The number of steps it takes to find the new reward increases when the multi-step information emission is replaced by one step. **D**. The performance is impaired only a little when the order of information is shuffled.

point and replace them with random noise (Figure 3B, right). Performance decreases monotonically as more replay steps are masked (Figure 3B, left). This proves that the information is incrementally unrolled as a sequence during replay. When we shuffle the order of the messages sent during replay, the performance is only slightly affected (Figure 3D), suggesting that the information may be sent in the form of independent packages rather than a whole sequence.

### 3.3 INFORMATION FLOW DURING REPLAY ENTAILS INFORMATION ABOUT CONTEXT AND ACTION PLAN

Thanks to the gray-box nature of AI, we can now extract the hidden states of each module and analyze the information inside. We assume that, the PFC initially has no idea that the reward location has switched from C1 to C2, but it changes its context encoding and gets to know this through replay. To test this, we train a Gaussian Naïve Bayes decoder to predict the reward location from the PFC's hidden states, and observe a gradual increase in decoding accuracy as the agent first encounters the new reward location, which remains high with repeated encounters (Figure 4A, middle). The analysis on HF hidden states has the same result (Figure 4A, left). This shows that replay helps identify the new context quickly. Additionally, another decoder trained on the messages sent from HF to PFC achieves high accuracy in predicting reward location. These results suggest that HF-PFC information emission helps update memory.

We also train a Ridge Classifier to predict actions from PFC hidden states, and find that decoding errors decrease significantly as the agent gets to the new checkpoint repeatedly. This shows that replay helps form future direction intentions (Figure 4B). It's important to note that the weights are fixed during the entire test period, so the model learns how to use replay to update memory and form plans to complete the flexible navigation task.

In the end, we investigate how replay guides the update of hidden states in the PFC using a "stop and scan" paradigm. When replay ends, we pause the real-time clock and allow the agent to explore the map randomly for 100 steps, recording the PFC's value output as its estimation of every location. Finally, we plot the values of different locations in the whole map (Figure 4C). As predicted, the high-value area shifts from S-C1-G to S-C2-G after the change of reward location (Figure 4E, left and right are for S-C and C-G respectively). We then convolve the value map with a Difference of Gaussian (DoG) filter. The filter has a positive peak at C2 and a negative peak at C1, which could

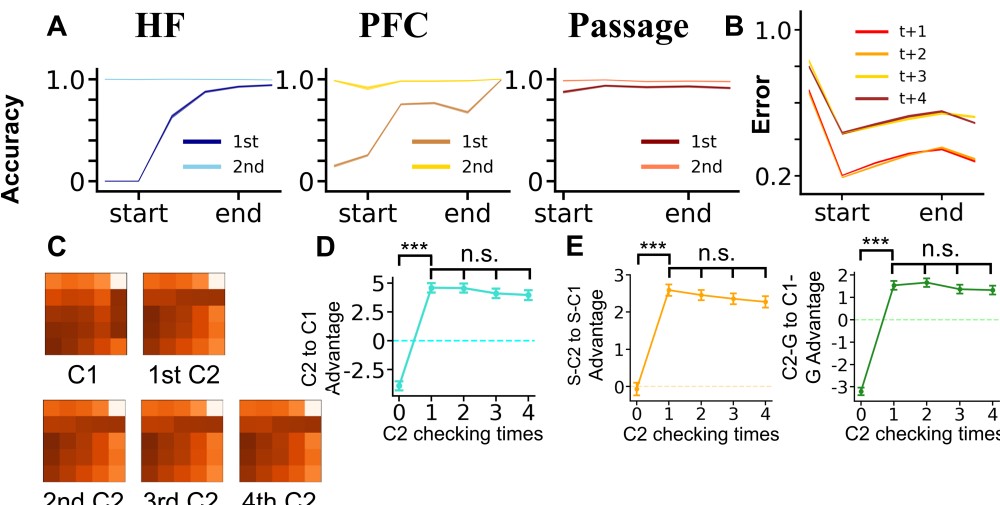

Figure 4: **A**. Decoding accuracy for the reward location by activities in HF (left), PFC (middle), and information passage (right), respectively. The X-axis is composed of three parts: the initial stage before replay, steps 1-4 during replay, and the output stage after replay. **B**. Decoding error for correct future actions from PFC activities after replay. Red, orange, yellow, and brown denote the first, second, third, and fourth future action following replay. In A-C, darker colors represent the agent's first getting the relocated reward, while lighter colors represent the second time. **C**. Value map produced through the "stop and scan" method. The C1 map represents the value before the checkpoint change, and the subsequent value maps are when the agent gets to C2 repeatedly. **D**. Value advantages of checkpoint 2 over checkpoint 1, calculated by convolving the value map with a Difference of Gaussian (DoG) filter. **E**. Left, value advantage calculated as the value of the path S-C2 minus the value of S-C1. Right, the value of the path C2-G minus the value of C1-G.

measure the relative advantage of the area around C1 to C2. The result that the advantage changes from negative to positive confirms the conclusion above (Figure 4D).

### 3.4 MANIFOLD ANALYSIS REVEALS THAT REPLAY FLOW BRIDGING BETWEEN CONTEXTS

To better visualize the dynamic changes in PFC activities, we perform a 3-dimensional principal component analysis (PCA) on PFC hidden states during the whole trajectory (Figure 5A). Each trajectory is composed of three stages: moving from the starting point to the checkpoint, getting the reward and generating replay, and moving from the checkpoint to the goal. For visualization, we only pick those stages that the animal finishes in four steps. Each step has a different color shown at the bottom. We connect the centers of the data point groups of each step to form an average trajectory. We use two distinct colors to differentiate activities during replay and the real movement. For example, in the left figure, we use red to denote the PFC hidden states during movement and green to denote hidden states during replay. The red continuous dots are separated by the green dots just like how a real trajectory is separated by replay. We use gradually darker colors in each trajectory to represent the gradually increased time steps, shown at the bottom. In the left and right figures, the reward is at a known point, and we can see the PFC hidden states constitute an orbit. In the middle figure, the agent first "thinks" the reward is at C1, then changes its mind and starts to explore, and finally finds C2, and the PFC hidden states settle on the new orbit. Replay connects two low-dimensional context subspaces. The activities during random exploration when the animal cannot find the reward are too dispersive to see clearly, so we don't show them here.

Then we analyze the dimensions of different subspaces spanned by PFC hidden states. The dimension is defined as the first index of PC whose accumulated explained variance (AEV) reaches 70%. Figure 5B is a representative AEV figure of PCs of PFC hidden states during replay when the agent finds the relocated reward. The dimension of this subspace is 2. Similarly, we show the subspace dimension of PFC hidden states only during replay (Figure 5C, left), or in the whole trajectory (Fig-

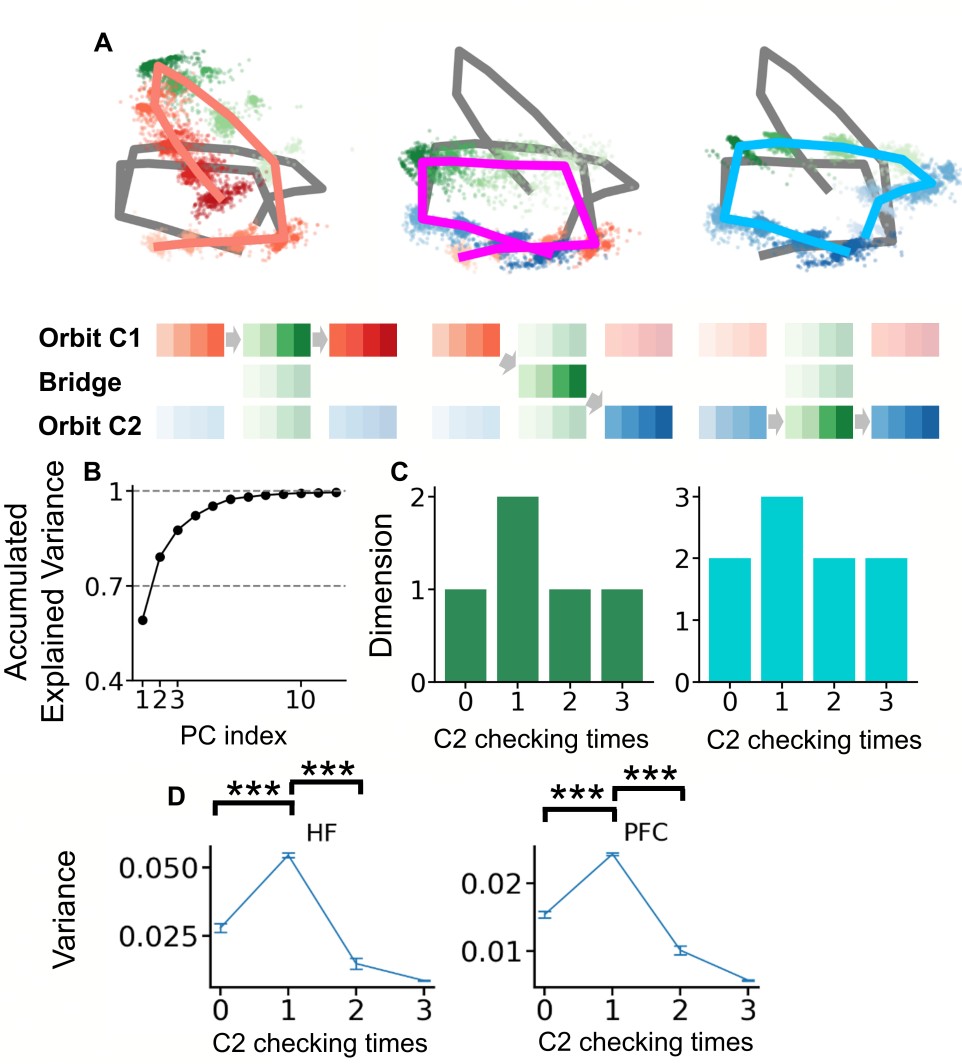

Figure 5: Manifold analysis reveals the context-switch process in detail. **A**. The 3D embedding of the "neural" manifold through dimension reduction of PFC activities when the small intermediate reward stays at C1 (left), the agent meets the relocated reward at C2 for the first time (middle), and the agent meets C2 for the second time (right). The highlighted trajectory represents the neural manifold at the corresponding stage, but the other trajectories are also plotted for comparison. **B**. AEV of different PCs of PFC activities during replay when the agent first finds the new reward. **C**. Left, the dimension of the PFC activities during replay (virtual experience) when the threshold for AEV is set to 70%. Right, the dimension of PFC activities during both movement and replay. The results are the same. **D**. The mean square distance of data points to their KNN centroids calculated from hidden states in HF and PFC during movement.

ure 5C, right). When the reward location changes, the all-state subspace dimension briefly increases to 3 and then drops back to 2, indicating a more complex subspace during the context switch. If we consider one "context orbit" as a 2-dimensional circle, then switching between these two orbits would produce a 3-dimensional curve, This is a rough and intuitive understanding.

We also evaluate the trajectory stability during the context switch. We first perform the K-Nearest Neighbour (KNN, $K = 20$) algorithm on hidden states during movement to find the centroids for

the data points. Then we calculate the mean square distance of data points from their KNN centroids (Figure 5D) as the variance of the hidden states. The variance increases briefly during the context switch and then drops back when the agent gets to C2 for a second time. $K = 8$ gives out the same result. This shows that the first encounter of the new reward is accompanied by an unstable and dispersive trajectory. And the trajectory becomes more stable in the following time.

# 4 DISCUSSION

In this article, we have demonstrated that biologically similar replay can emerge from the interaction between the HF module and PFC module within a task-optimization framework. We show that replay in our model is actually a form of multi-step information emission from HF to PFC. Then we show that during replay, the context is updated both in HF and PFC, and the cognitive map and future plan are updated in PFC. Therefore we describe how replay helps learning. At last, we plot the manifold to give a more intuitive understanding. See Appendix A.3 for reproducibility results.

## 4.1 PLAUSIBILITY

**Conditions** Our two sufficient conditions (1) are inspired by neuroscientific research. For Condition (1), studies have shown that blocking replay impairs spatial learning (Jadhav et al., 2012; Girardeau et al., 2009; Ego-Stengel & Wilson, 2010). The presence of task-focused replay can predict the accuracy of following decisions (Ólafsdóttir et al., 2017). Replay is modulated by reward (Ambrose et al., 2016; Liu et al., 2021; Michon et al., 2019), and is associated with reward-guided updates of internal models based on past experiences (Carey et al., 2019).

For condition (2), the interaction between HF and PFC during replay is supported by a lot of evidence. For example, replay is accompanied by sharp-wave ripples that synchronize the wave phases of HF and PFC (Tang et al., 2017). Replay is thought to be the substrate of the continuous memory transition from the hippocampus to the neocortex, namely memory consolidation(Kumaran et al., 2016). Recent computational models also reveal that the communication between the world model and policy network could help decision-making (Jensen et al., 2024).

However, there is some evidence that does not support this hypothesis as a generating condition for replay. For example, Kaefer et al. (2020) reported PFC replay that is strongly associated with rule-switching performance but is independent of HF replay. As we do not measure PFC replay, it may be that not replay but other PFC activities in our model. Future work could design experiments to make this clear.

**The segregation of HF and PFC during movement** The notion that the HF and PFC are entirely segregated during movement may seem somewhat implausible. We acknowledge that this highlights a limitation of our model's fidelity to realistic biological settings. However, we focus on replay which occurs only at rest. During mobility, the HF and PFC could communicate with each other not by replay but by theta waves, which are thought to be mainly driven by external signals and involve only nearby place cells' firing, rather than a global sequence. Allowing the two modules to communicate during movement might add unnecessary details less relevant to the problem we want to analyze. However, it would also be exciting to see how adding biological details could still produce the phenomena we want and how different factors interact with each other.

**The difference of PFC from the striatum** Previous RL theories for animal behaviors also emphasize the striatum as the center for model-free learning. The dorsal lateral striatum (DLS) and the ventral lateral striatum (VLS) correspond to the actor and critic parts in the classical Actor-Critic algorithm. The dopamine serving as TD signal is calculated in the ventral tegmental area (VTA) and drives synaptic plasticity in the whole network (Montague, 1999). Does this role of striatum overlap with PFC? Wang et al. (2018) proposes a meta-reinforcement learning model where PFC is the inner-loop "fast" learner and the dopamine system is the "slow" learner. We think it's also useful to consider the training process involved weight changes as a kind of dopamine learning, and later context identifying and following decision-making involved hidden state changes as PFC learning.

## 4.2 HOW REPLAY PROMOTES LEARNING

The classical view posits that replay aids value-based reinforcement learning (Liu et al., 2021; Foster & Wilson, 2006; Mattar & Daw, 2018; Ambrose et al., 2016), and our model provides a concrete implementation supporting this view. In our model, cognitive map modulation in the PFC changes as a result of replay and then stabilizes (Figure 4). A value-modulated cognitive map in the PFC is reminiscent of value and task structure representations in the orbitofrontal cortex (OFC) (**?**;Zhou et al., 2019; Padoa-Schioppa & Assad, 2006), a part of PFC. Our work suggests these representations could combine as a value function (V function) to guide decision-making. Our PFC cognitive map is formed with the help of HF, which also aligns with previous findings (Mızrak et al., 2021).

Despite those model-free components, our model should still be classified as model-based. The main difference from AI model-based methods is that they sample from the distribution learned by the world model and use these samples to improve the policy. The advantage is that it could provide gradients for the sampling process and have the gradients of the reward loss directly back-propagated to the policy network. Our method, however, doesn't design how the world model should help the policy network. The replay in our model is similar to a sampling process and maybe gradients are propagated through this process but future work needs to make this clearer. Recently Jensen et al. (2024) explores the model-based explanation of replay, but they also have an explicit design that the samples from the world model are directly taken as input to the policy network.

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

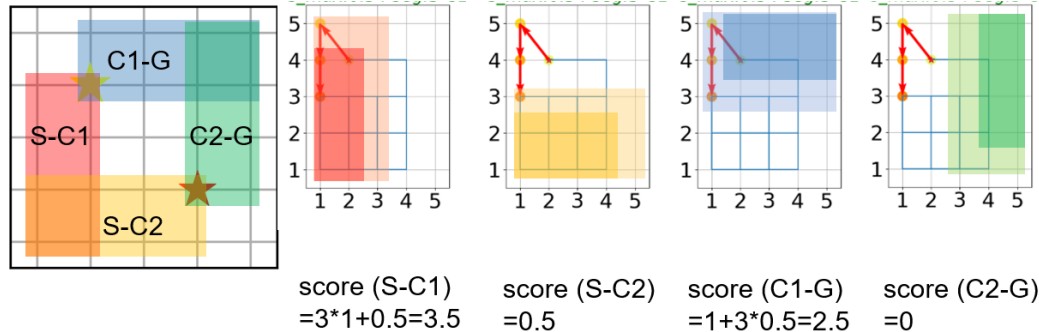

score (S-C1)     score (S-C2)     score (C1-G)     score (C2-G)
=3*1+0.5=3.5     =0.5             =1+3*0.5=2.5     =0

Normalized_score = [3.5/6.5, 0.5/6.5, 2.5/6.5, 0/6.5] = [0.54, 0.08, 0.38, 0]
Attribution = S-C1

Figure 6: Calculation of replay distribution

# A    APPENDIX

## A.1    CALCULATION OF REPLAY DISTRIBUTION

Replay is defined as the sequential activation of place cells. Therefore, a replay trajectory would represent a path in the room. We determine the attribution of replay by calculating the overlapping of our replay with different parts of the room (Figure 6 left). Equal overlapping would cause exclusion of that trajectory. Finally, the numbers of replay trajectories representing each part of the room are normalized to get the spatial distribution of replay, namely "replay distribution".

The rule we use is a little more complicated. Each part of the room is divided into two segments with scores that equal to 1 and 0.5 respectively. subsection A.1 provides a concrete example. The red trajectory is one replay.

## A.2    ANNOTATION FOR FIGURES

All error bars indicate $\pm 1$ standard error.

In Figure 4, for all decoding tasks, 80% of the data is used for training and 20% for testing. The Gaussian Naïve Bayes and Ridge Regression methods are used to decode the reward location and future directions respectively. Each step has an independent decoder, and they do not share training sets.

In Figure 5C, measurements are repeated for 10 times with different seeds. The results are the same.

## A.3    REPRODUCIBILITY

We repeat the experiments across different seeds, and the result has been presented in the main text. Then we retrain the RL across different seeds and different numbers of replay steps, the result is shown in Figure 7.

## A.4    TRAINING CURVE OF MULTIPLE AGENTS

The RL agents are tested every 5000 steps. The training curves are shown in Figure 8. The model without HF clearly receives fewer rewards than the other models.

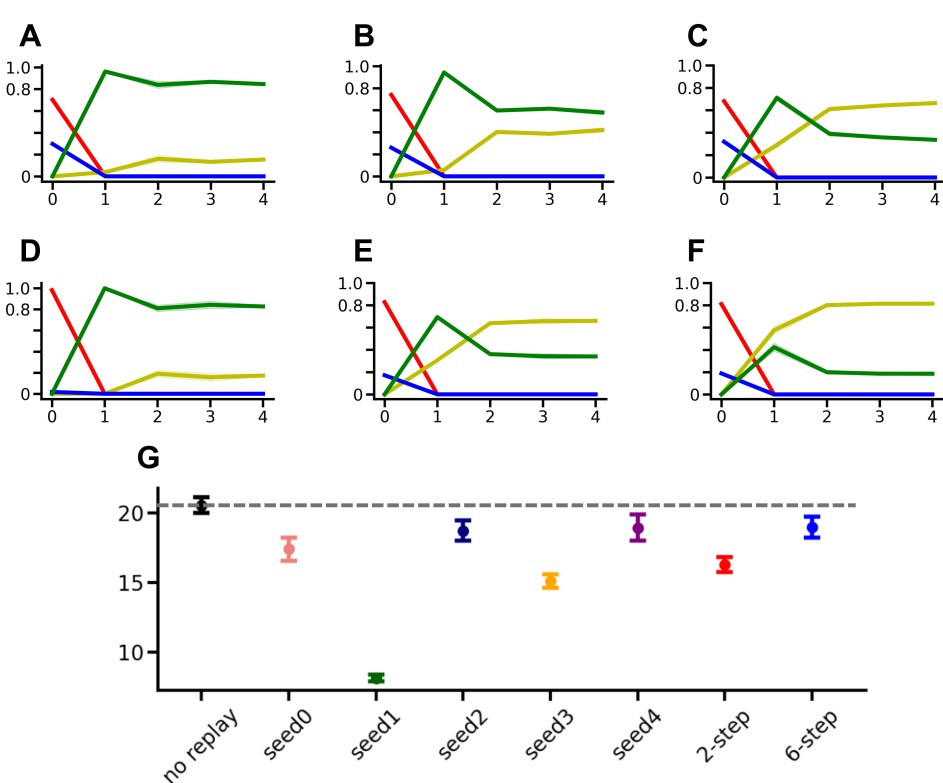

Figure 7: **A-F**. Changes of replay distribution across different seeds or different numbers of replay steps, like in Figure 2E. They have the same trends that the replay of green parts first increases and then decreases, and yellow parts keep increasing. **A**. Seed = 1. **B**. Seed = 2. **C**. Seed = 3. **D**. Seed = 4. **E**. Number of replay steps = 2. **F**. Number of replay steps = 6. **G**. Exploration steps as in Figure 3C. The model without replay suffers from lower exploration efficiency.

756
757
758
759
760
761
762
763
764
765
766
767
768
769
770
771
772
773
774
775
776
777
778
779
780
781
782
783
784
785
786
787
788
789
790
791
792
793
794
795
796
797
798
799
800
801
802
803
804
805
806
807
808
809

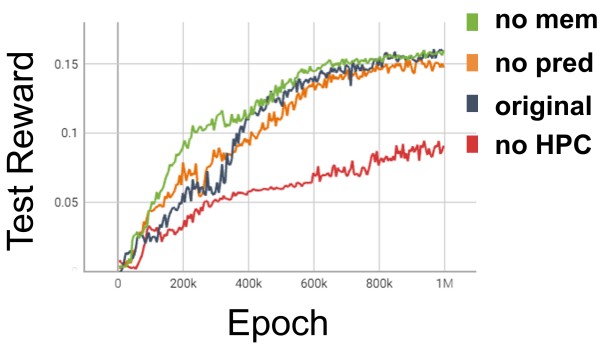

Figure 8: Training curve of different models

