# OpenReview forum: "Brain-Like Replay Naturally Emerges in Reinforcement Learning Agents"
_ICLR.cc/2025/Conference — Submitted to ICLR 2025_

### Official Review · Reviewer_DDUc · 2024-10-27

**Soundness:** 2
**Presentation:** 1
**Contribution:** 2
**Rating:** 3
**Confidence:** 3

**Summary:**

This paper proposes that replay emerges naturally during reinforcement learning as a result of communication between a ‘hippocampal’ world model and ‘prefrontal’ policy. To demonstrate this, the authors set up a network in which communication between the world model and policy occur only during ‘rest’ periods after a reward has been received. They test this on a simple animal-inspired task in which the path to acquiring reward changes. They show how the place cell activation changes over time to reflect the updated position, and that this roughly aligns with the dynamics of previously reported rodent replay. They further show that intervening on the communication channel from the world model to the policy disrupts the behavior, and that multiple steps of communication between the world model and policy appears to yield better behavior.

**Strengths:**

- The premise that replay may emerge naturally as a result of a broader reinforcement learning process is interesting.
- The hypothesis that communication between the world model and policy is important for generating replay is interesting.
- The connection to animal data is interesting.

**Weaknesses:**

- The presentation, experiments, and analyses make it difficult to discern what to actually conclude about their hypothesis and premise, and I am not convinced to a degree that would be necessary for publication. See detailed questions below.

- I would want much more detailed and convincing demonstration of the actual replay, perhaps closer to the neuroscientific representation such as in Carr et al. 2011, ‘Hippocampal replay in the awake state: a potential substrate for memory consolidation and retrieval’ (e.g. Fig 1), and in more straightforward replay settings (such as a linear track or W maze).

**Questions:**

1) In Figure 1, where does the replay trajectory in F come from? I realize it is a schematic, but it is somewhat confusing how to interpret it.

2) In equations 2, 3, should I_replay be named something else, such as I_rest, otherwise it is sort of asserting the conclusion in the definition?

3) Does the r in W_{h, r} stand for replay? Or reward? I’m somewhat confused by what is going on there.

4) What is the axis of 2E? What is the relationship between the x-axis of 2C and 2E? What is the color scheme in 2E? What are the vertical dashed lines in 2D? Why is step count negative in 2A (and why are the y-ticks covered up)?

5) ‘C2 checking times’ are never explicitly introduced as a metric, and it’s unclear what the means as an x-axis in 2E.

6) I don’t know exactly what it means for the results in Figure 2 to prove that conditions in equation 1 are sufficient to generate replay. What are alternatives outcomes that could have happened that might not have demonstrated this?

7) In 4A, what is ‘1st’ and ‘2nd’?

8) Is replay actually necessary for the results in Figs 3 and 4, or is it just a quirk of how this model was trained. In particular, this model was trained using RL in a setting where after reward was received, the communication channel opens up, so it will naturally find some way to use that communication channel. So disrupting that channel should disrupt behavior, but I'm not certain that is a particularly deep conclusion.

9) My understanding is that this model is structurally designed to produce place cell activations. Thus, when it is unlinked from environmental input, and the ‘PFC’ activity yields place cell activations in ‘HF’, is there any alternative? Should the place cells just be silent? This seems like a fairly weak way to assert that replay emerges. Can we see more of an emphasis on the actual spatial trajectories?

10) If the weights are fixed, then what is the role of replay? This is suggesting a role for replay during in-context updating, but I’m a little confused how that relates to the idea of using replay for weight updates and Experience Replay. I was really excited at the beginning of the paper that it would show how replay-for-updating-synaptic-weights could emerge, but I don't see any discussion of that.

11) Have any animal experiments been published that show that blocking communication between PFC and HF blocks replay? If so, it should be made extremely clear in the text. This should be a highly doable experiment using optogenetics.

12) When testing the necessity of HF to PFC channel: is signal only replaced during replay periods? It’s not clear to me how disrupting the transmission from HF to PFC actually disrupts PFC, in 3A. Is it just because the state of PFC is disrupted, and because that state is disrupted, then the selected actions are also disrupted?

13) I don’t follow what the implication is that shuffling the messages during replay has minimal effect. What is an example of an ‘independent package’?

---

### Official Review · Reviewer_oWjc · 2024-10-30

**Soundness:** 3
**Presentation:** 2
**Contribution:** 3
**Rating:** 3
**Confidence:** 2

**Summary:**

This paper presents a novel, neuroscience-inspired model that successfully demonstrates the replay phenomenon without relying on hardcoded rules. However, the experiments are limited to a specific setting, and the paper does not provide justifications that the replay mechanism improves exploration across broader RL algorithms and tasks.

**Strengths:**

- This paper introduces a neuroscience-inspired model that simulates the replay phenomenon without predefined or hardcoded replay rules, which is interesting, innovative and aligns with biological replay mechanisms observed in the brain.

- The authors demonstrate the functional importance of replay by drawing parallels with the biological experiment in [1]. The results show that its RL agent’s replay patterns match the replay distribution and adaptive pathfinding seen in rodents. The abolation studies also show how disrupting the replay signal impairs the agent's exploration efficiency.


**References**:

[1] Hideyoshi Igata, Yuji Ikegaya, and Takuya Sasaki. Prioritized experience replays on a hippocampal predictive map for learning. Proceedings of the National Academy of Sciences

**Weaknesses:**

- The paper should contain a background section that describes the overall problem formulation along with some preliminary knowledge. This would help improve the readability of the main technical section.

- The paper does not sufficiently demonstrate how the replay phenomenon would enhance exploration in broader RL settings. The experiments are restricted to a specific grid-world task, limiting the ability to assess the model's general applicability. Additional downstream tasks or diverse environments would help clarify the robustness and transferability of the proposed methods.

- Although the authors claim the model is modular, there is limited evidence showing how this modular framework could integrate effectively with mainstream model-free (value-based) algorithms. If would be helpful if the authors can discuss how and HF and PFC could improve some standard RL algorithms.

**Questions:**

- Regarding weakness 2, I am not sure what the authors mean by "modular" in the abstract. I assume this means the proposed model is a general framework on its own and can be integrated with other blocks. If the authors meant something else, please provide clarifications.

- The paper classifies the approach as "model-based," primarily due to the role of the HF as a "world model". Could the authors provide empirical evidence showing HF’s accuracy in predicting next states or rewards, demonstrating its role as an explicit model?


I am happy to raise the score if the points in the weaknesses and questions can be addressed.

---

### Official Review · Reviewer_Qi75 · 2024-11-01

**Soundness:** 2
**Presentation:** 1
**Contribution:** 1
**Rating:** 3
**Confidence:** 2

**Summary:**

The authors propose a biologically-inspired model-based RL architecture for reinforcement learning. They train this network using a two-stage process to learn a world model and policy separately. They show that this architecture is capable of 'generating replay' organically from reward-based learning. They demonstrate using a simple toy setting that this architecture is capable of adaptation to changing reward structure without parameter modification. They also perform limited interpretability studies on the information being passed from the world model to the policy.

**Strengths:**

Originality: Uncertain. I am not familiar with other work.

Clarity: Poor. The authors do not define key terms such as ‘replay’ and ‘place cell’. There are also multiple instances where methodology is not described with enough precision to fully understand the experiment that has been conducted. Lastly, the paper uses many jargon terms which are presumably familiar to neuroscience researchers, but which are unfamiliar for general machine learning researchers. Overall, the paper is difficult to understand and would benefit from significant rewriting to make it more accessible to the broader ML community.

Quality: Uncertain. I am not able to accurately assess the quality of the paper at the moment, due to significant conceptual confusions regarding the research question under investigation, the  experimental methodology used, as well as the broader significance of the result.

Significance: Poor. The stated impact of the paper seems to be that agents can ‘generate replay’, and this may be significant to neuroscientists, but the relevance to the broader machine learning / RL community is unclear. As a machine learning researcher, the part I found most interesting was the demonstration that the network was capable of adapting to a different reward location at test-time. Adopting this framing, I find the experiments or results extremely lacking, since the task is extremely toyish and there is no comparison to baselines.

**Weaknesses:**

The paper’s central result is that the proposed experimental design is capable of ‘generating replay’ organically from reward-based learning. However, it is never concretely defined as to what ‘replay’ is or how replay can be ‘generated’. A clear mathematical definition would be useful. Furthermore, it is not explained what the significance of ‘generating replay’ in this fashion is. Lastly, it is not clear whether these claims generalise to a broader RL setting, since the task is extremely toy-ish.

The figures are not well explained. In Figure 1A and 1E, what does “C2 checking time” mean? In Figure 1C, how are the boundaries between the different stages determined? In Figure 1D, what is ‘distance’? In Figure 3B, what does it mean to mask a replay step? In Figure 3C, what does it mean to replace the multi-step emission with one step? In Figure 3D, what does it mean to shuffle the order of information?

In Section 3.4, it is unclear what conclusions should be drawn from the manifold analysis. What are the takeaways and why are they important?

**Questions:**

Please provide definitions for the following terms (mathematically, where possible): ‘place cell’, ‘replay’, ‘replay step’, ‘phase-lock’, ‘one-step information emission’.

The RL task used in all experiments is not sufficiently explained. While the high-level intent is clear, low-level details are lacking. Please include a precise definition of the environment’s transition dynamics and reward structure.

In Section 3.1, when attributing replay trajectories to one of the four paths, what happens to trajectories that do not clearly pass through / near either of the checkpoints?

In Section 3.3, the authors write that they “pause the clock and allow the agent to explore the map randomly for 100 seconds”, but it is highly unclear what this actually means. What are the inputs and outputs of the network during this time?

---

### Official Review · Reviewer_91TP · 2024-11-07

**Soundness:** 2
**Presentation:** 3
**Contribution:** 2
**Rating:** 6
**Confidence:** 2

**Summary:**

Replay has an important role in the brain, and experience replay in RL seems to be important and beneficial in RL. However, experience replay in RL is a hardcoded and predefined procedure in the sense that it is explicitly implemented rather than being an emergent phenomenon. This paper is rather exploratory in that it takes inspiration from replay in the brain and develops a model based off of it to investigate the emergence of replay in RL agents without explicitly programming in experience replay. Inspired by the brain, they propose/hypothesize two conditions for replay to motivate their model. They implement a modular neural network with different parts serving as analogs to different parts of the brain. This model exhibits replay and they then study replay within this model.

**Strengths:**

This paper is interesting, with a large amount of neuroscience inspiration and guidance into developing the model. The specifics of the model do not look easy to implement, so the emergence of replay is certainly interesting. This paper seems exploratory, and in such papers that aim to model some natural phenomenon in the brain it is naturally difficult to ascertain the validity and accuracy of the model. This is fine, as these kinds of works are just steps towards future investigations. The design of the experiments seem well done and I do believe that the claims of the paper are consistent with the experiments.

**Weaknesses:**

They do evaluate their model on essentially a single environment, and it does call into question the degree to which the results and the model generalize for studying replay. They do use an environment that is based on prior research, which is good. Nonetheless, the generality of the results would be somewhat a concern of mine.

**Questions:**

- "There are several works on how to design more sophisticated experience replay mechanisms, like hyperparameter tuning (Fedus et al., 2020)". I don't think Fedus et al. 2020 do this. They do investigate what happens under different replay buffers sizes and update rates. But beyond that, there is not much.
- The citation "(Van de Ven & Tolias, 2018)". The "V" in "Van" should be lower case.

---

### Meta-Review · Area_Chair_eSjA · 2024-12-20

**Metareview:**

This paper explores the emergence of replay in reinforcement learning (RL) agents, drawing inspiration from the replay phenomenon observed in the brain. The authors propose a biologically-inspired model-based RL architecture and demonstrate that replay can emerge naturally without explicit programming.

Strengths
-----------
- **Novel perspective:** The paper offers a fresh perspective on replay in RL by investigating its emergence as a natural phenomenon rather than a hardcoded mechanism.
- **Interpretability:** The authors conduct analyses to understand the information flow within the model and the role of replay in adaptation to changing reward structures.

Weaknesses
--------------
- **Limited generalizability:** The experiments are primarily conducted in a single, albeit well-established, environment, raising concerns about the generalizability of the findings to other RL tasks and algorithms.

- **Lack of clarity:** Reviewers found the paper difficult to understand in parts, particularly regarding the definition and significance of "replay" and the explanation of certain figures and experimental procedures.

- **Limited significance for RL:** While the emergence of replay is interesting from a neuroscience perspective, its practical implications for improving RL algorithms remain unclear.

This paper presents an interesting exploration of the emergence of replay in RL agents, drawing valuable insights from neuroscience. However, it needs further development to address concerns about generalizability, clarity, and relevance to the broader RL community.

**Additional Comments On Reviewer Discussion:**

The authors have not provided a rebuttal and no discussion has been done.

---

### Decision · Program_Chairs · 2025-01-22

Reject